# Dietary Shifts since COVID-19: A Study of Racial Differences

**DOI:** 10.3390/nu16183164

**Published:** 2024-09-19

**Authors:** Lillie Monroe-Lord, Azam Ardakani, Phronie Jackson, Elmira Asongwed, Xuejing Duan, Amy Schweitzer, Tia Jeffery, Tiffany Johnson-Largent, Elgloria Harrison

**Affiliations:** 1College of Agriculture, Urban Sustainability and Environmental Sciences, University of the District of Columbia, Washington, DC 20008, USA; azam.ardakani@udc.edu (A.A.); phronie.jackson@udc.edu (P.J.); elmira.asongwed@udc.edu (E.A.); amy.schweitzer@udc.edu (A.S.); tia.jeffery@udc.edu (T.J.); tiffany.johnsonlarge@udc.edu (T.J.-L.); 2Data Analytics, McDaniel College, College Hill, Westminster, MD 21157, USA; xuejing937@gmail.com; 3Lehman College, City University of New York, Bronx, NY 10468, USA; elgloria.harrison@lehman.cuny.edu

**Keywords:** COVID-19, racial differences, dietary changes, nutritional vulnerability

## Abstract

Background: The COVID-19 pandemic has fundamentally changed the quality and quantity of people’s food consumption. Objective: This study aimed to explore the dietary shifts among different racial groups resulting from the COVID-19 pandemic, focusing on changes in consumption across various food categories. Methods: This cross-sectional study included a sample of 10,050 urban residents aged 40–100 years across the United States. Dietary patterns among African American, Asian, Hispanic, and White populations were assessed before and since the pandemic (retrospective condition) using the Dietary Screening Tool (DST). The DST investigates consumption trends in food groups aligned with the MyPlate guidelines, plus fat, sugar, and sweet (FSS) intake and processed meats. Results: This study found significant shifts in food consumption patterns among racial groups since COVID-19. The data indicate that African American individuals largely reduced their consumption of several food groups compared to White individuals, with a 43% decrease in processed meats, 42% in dairy, 36% in lean protein, 21% in fruit, 17% in grains, and 15% in FSS, although their vegetable consumption did not significantly decrease. African American individuals also consumed 66% less processed meat, 57% less dairy, and 30% less lean protein in comparison to Asian individuals. Hispanic individuals also showed a tendency to reduce their consumption more than White individuals, with a 34% decrease in dairy, 28% in vegetables, and 24% in fruit. In contrast, Asian individuals consumed 37% less FSS and 34% less grains than White individuals. Additionally, when compared to Asian individuals, Hispanic individuals consumed 49% less dairy and 47% less processed meat. The findings also revealed that African American individuals were the most nutritionally vulnerable group since the pandemic. Specifically, they were 38% and 35% more likely to be considered at nutritional risk than Asians and White participants, respectively. Conclusions: These findings illuminate the considerable dietary shifts induced by the COVID-19 pandemic and emphasize the critical need to address the racial disparities in nutritional vulnerability and public health policy.

## 1. Introduction

Due to lockdowns and pandemic-related restrictions, the COVID-19 pandemic substantially changed people’s dietary habits worldwide [1,2]. Various studies have been conducted to understand how food consumption patterns have changed due to the pandemic [3,4,5,6,7,8,9,10,11,12]. Interestingly, the findings from these studies have been somewhat contradictory [2]. On the one hand, some research has indicated that certain individuals adopted healthier eating practices during the COVID-19 lockdowns, as they had more time to cook healthy meals at home and decreased their fast food consumption [5,13,14,15]. However, on the other hand, some studies have revealed an increase in snacking and meal frequency and the consumption of unhealthy food options and alcohol during the lockdowns, along with a corresponding decrease in fruit and vegetable consumption [13,16,17].

To compound the issues surrounding dietary habits, previous research has highlighted disparities in dietary patterns and health outcomes among different racial and ethnic groups [18]. Specifically, studies have shown that different racial populations often exhibit unique dietary preferences, cultural practices, and socioeconomic factors that influence their food choices and eating behaviors [19,20]. For instance, a study by Tao, Liu, and Ngyuen (2022) revealed that from 2011 to 2018, the non-Hispanic White group exhibited a notable reduction in their overall Healthy Eating Index 2015 (HEI-2015) scores, while this trend was not observed among other racial/ethnic groups. However, this trend has been disrupted since COVID-19 [21]. As such findings show, it is essential to examine not only how broad dietary patterns have been impacted by the COVID-19 pandemic but also how dietary changes have manifested in terms of racial/ethnic groups.

The COVID-19 pandemic was characterized by alarming disparities in infection rates and outcomes across different racial and ethnic groups, with racism, other medical conditions, types of work, living locations, access to healthcare, hospitalization rates, and comorbidities associated with COVID-19 all contributing to these inequities [22,23,24]. Studies have shown that African American individuals, in particular, were disproportionately affected by the virus, experiencing higher rates of infection, hospitalization, and mortality [24]. According to COVID-19 disparities from a biomedical lens, individuals with diabetes mellitus, hypertension, renal disease, and obesity are at a higher risk of experiencing severe COVID-19 symptoms and face a higher likelihood of mortality; these diseases have a high prevalence among African American individuals [25,26]. Additionally, non-Hispanic African American patients had 2.7 times higher odds of hospitalization compared to non-Hispanic White patients [23]. Similarly, Latino populations have also faced elevated risks and adverse outcomes of COVID-19 [27]. Another study showed that COVID-19 disrupted food shopping habits as well. Factors such as reduced store hours, food shortages, and concerns about infection contributed to these changes. Findings from a nationwide survey, for instance, revealed that although Asian households did not experience significantly higher levels of food insecurity compared to White households in late April 2020, they displayed a greater tendency to avoid shopping for food due to fear of exposure and, additionally, encountered transportation challenges when purchasing groceries [28].

Despite extensive research into the impact of the COVID-19 pandemic on dietary patterns, there remains a significant gap in our understanding of how these changes have manifested across different racial and ethnic groups within the American population. Most studies have broadly examined dietary shifts without delving into the nuanced differences among diverse cultural backgrounds. Recognizing this gap, our study endeavored to fill this void by employing a unique approach. Specifically, it aimed to illuminate the intricate dietary changes among diverse racial groups, including African American, Asian, Hispanic, and White populations, aged 40–100 years old, residing within the United States, across all MyPlate food categories. This effort is crucial in understanding the multifaceted impact of COVID-19 on dietary changes.

This research had two main goals. First, this study aimed to explore how COVID-19 impacted the dietary changes of various racial groups, an area that has not been extensively researched. Second, it intended to understand whether certain racial groups faced greater nutritional vulnerability due to the pandemic. Distinguishing itself from prior research, this study examined dietary changes in the later stages of the pandemic, a pivotal distinction from the predominantly early-stage analyses that have thus far been conducted. Such an understanding can inform the development of culturally sensitive interventions and policies to mitigate adverse dietary changes and promote healthy eating practices. Furthermore, identifying race-specific dietary changes during the pandemic can provide valuable information for healthcare professionals, policymakers, and community organizations striving to address health disparities and improve population health outcomes.

## 2. Methods

### 2.1. Design, Participants, and Procedure

Focusing on COVID-19’s impacts on dietary changes, this cross-sectional study examined vulnerable urban populations across various racial groups. The research protocol received approval from the Institutional Review Board at the University of the District of Columbia. The study encompassed 10,050 participants aged 40 to 100 years who were enlisted between 9 August and 15 September 2020. To ensure diverse representation from urban areas across all four census regions in the United States, we used Qualtrics (2020) to manage recruitment through an online survey panel [29]. These urban populations were chosen because they faced a higher risk of severe COVID-19 symptoms or complications. The United States Census Bureau (2013) defines urban cities as places with a population exceeding 50,000 residents [30].

### 2.2. Demographic Characteristics

Participants’ demographic information was gathered, including their gender (male, female), a combined classification of their race (White, Asian, Black, Hispanic), and their ethnicity (Hispanic, non-Hispanic). Alongside race and ethnicity classifications, a minority status variable was established, categorized as “yes” for non-White (non-Hispanic) participants and “no” otherwise.

### 2.3. Nutritional Assessment

For the assessment of participants’ nutritional status, the 25-item short-form Dietary Screening Tool (DST) was employed. This tool was developed and validated for usage in older and middle-aged populations [31,32,33]. The DST has demonstrated good internal consistency, with Cronback’s alpha coefficients ranging from 0.78 to 0.85 in previous studies [31,32,33]. Additionally, the DST has shown strong constrict validity, correlating well with other established dietary assessment methods. The DST employs a range of response options, including a range of food groups and dietary behaviors, with each question being assigned a score spanning from zero to eight. The total score of the DST spans from 0 to 100, with a higher score aligning with a healthier diet. For instance, one of the questions in the DST asks, “How often do you eat carrots, sweet potatoes, broccoli, or spinach?”, with response options that range from “Never” (0 points) to “Three or more times a week” (8 points).

The DST was administered twice in retrospect, before and following the COVID-19 pandemic, enabling the measurement of nutritional changes over this timeframe. The DST questions were aggregated into seven food groups, encompassing all MyPlate food categories: fruits (scored from 0 to 10), grains (scored from 0 to 15), vegetables (scored from 0 to 15), lean protein (scored from 0 to 10), and dairy (scored from 0 to 10). Additionally, the groups include fats, sugars, and sweets (FSS) (scored from 0 to 25) and processed meats (scored from 0 to 10). MyPlate is a visual guide created by the United States Department of Agriculture to help Americans understand the proportions of different food groups that should be consumed at each meal. It emphasizes the importance of including all five food groups as part of a healthy eating pattern [34]. Additionally, we included FSS and processed meats to capture the dietary components typically associated with increased health risks when consumed in excess. The FSS category can be part of a healthy diet in moderation but is often overconsumed and contributes to unhealthy dietary patterns. This grouping provides a simplified yet effective means for quantifying the consumption of these items, providing insights into dietary habits that may contribute to nutritional vulnerabilities or chronic health conditions [35]. The inclusion of processed meats as a separate category addresses the growing body of evidence linking their consumption to adverse health outcomes, including heart disease and certain cancers, offering a more comprehensive assessment of the dietary risks [36].

In the first approach, the mean and standard deviation of each food group’s consumption and the mean percentage change were calculated for each food group’s consumption before and after COVID-19. In the second approach, the participants were categorized into three groups based on the changes in their food intake since the pandemic: decreased consumption, no change, and increased consumption. In addition, each participant received a total DST score. Based on the total scores, the participants were categorized into three distinct groups of nutritional vulnerability: “at risk”, “possible risk”, and “not at risk”. The participants with DST scores lower than 60 fell within the “at risk” category, those with scores between 60 and 75 were classified as “possible risk”, and those with scores exceeding 75 were placed in the “not at risk” category. This categorization served as a valuable indicator of the nutritional status of the participants, enabling a detailed analysis of their level of dietary risk [31,33].

### 2.4. Software and Statistical Tests

Data analysis was performed using the SAS 9.4 software (SAS Institute, Cary, NC, USA). Descriptive statistics included the mean and standard deviation (SD) for continuous variables and the frequency and percentage for categorical variables. A paired-sample *t*-test was used to assess the differences in food items before and after the COVID-19 pandemic. The chi-square test examined the associations between categorical variables. Binary logistic regression models employing the MyPlate food items as response variables were utilized, with Fisher’s scoring optimization for modeling. These models predicted the likelihood of decreased food consumption since the pandemic while considering the explanatory variables gender, race, age, income, and education, alongside providing odds ratio (OR) estimates. Another binary logistic regression model used predictor variables to predict the probability of belonging to a group at higher nutritional risk since COVID-19. The response variable had two levels: participants without nutritional risk and those considered nutritionally vulnerable. Nutritional vulnerability was determined by assessing whether the participants had become “at risk” since COVID-19 compared with beforehand. The participants who were at risk before and remained at risk since COVID-19 were not classified as “nutritionally vulnerable”, whereas those who shifted from a possible risk to being at risk were categorized as “nutritionally vulnerable”. Statistical significance was defined as *p* < 0.05.

## 3. Results

### 3.1. Demographic Characteristics

The participants’ characteristics are presented in Table 1. The gender distribution shows that 42.6% were male and 57.4% were female. Regarding age, the majority fell within the 61–80 range (58.9%), followed by 40–60 years (38.5%), with a smaller percentage in the 81–100 category (2.6%). In terms of ethnicity/race, the largest group was White (74.5%), followed by African American (14%), Asian (7.1%), and Hispanic (4.3%). Regarding education, 16.3% had less than a high school education, 32% had some college education, and the majority (51.7%) held a college degree or higher. In terms of annual income, the distribution was as follows: 16.1% received less than USD 25,000, 23.3% fell within the USD 25,000–49,999 range, 34% within the USD 50,000–99,999 range, and 26.6% had an income of USD 100,000 or more.

### 3.2. Changes in Dietary Consumption by Race

Table 2 provides an overview of the changes in dietary consumption among various racial groups categorized by food items. For fruit consumption, all racial groups displayed significantly decreased consumption since the pandemic (*p* < 0.001), with the most significant decline observed in the Hispanic group (−7.16%), followed by the African American (−4.77%), White (−3.97%), and Asian (−3.41%) groups. In terms of grains, a significant reduction in consumption was observed across all groups (*p* < 0.001). The largest decrease was observed in the Asian group (−9.39%), followed by the Hispanic (−8.01%), African American (−7.54%), and White (−7.16%) groups. Regarding vegetables, only the White participants displayed a significant increase in consumption (0.41%, *p* < 0.001), while that of the other groups remained relatively stable. For lean protein, the African American and White groups showed a significant decrease in consumption (−2.76% and −0.98%, respectively). Dairy consumption declined significantly across three racial groups, with the largest decrease in the Hispanic group (−4.34%), followed by the African American (−3.44%) and White (−0.85%) groups. The mean FSS consumption score increased significantly in all four racial groups, indicating healthier habits or less FSS consumption since COVID-19. In terms of processed meats, the African American group displayed a slight but insignificant increase in consumption (1.62%). In contrast, the White group showed a minor and significant decrease (−0.67%), and the other groups remained relatively stable. The Hispanic group showed the highest increase in the mean DST score (5.37%), reflecting healthier habits, followed by the Asian (4.64%), African American (4.56%), and White (3.18%) groups.

Statistically significant racial disparities were evident in the shifts in dietary habits since the COVID-19 pandemic (*p* < 0.001). Across all seven food categories examined, FSS saw the largest trend of reduced consumption across all racial groups. Specifically, 45.8% of Asian participants, 44.8% of Hispanic participants, 42.7% of African American participants, and 37.8% of White participants reported a decrease in FSS consumption since the pandemic. Conversely, processed meats exhibited the lowest reduction in consumption across all racial groups compared with the other food categories. Approximately 20.4% of African American participants, 19.1% of Hispanic participants, and 13% of White and Asian participants reported a decrease in their intake of processed meats. Hispanic participants reported a 37.3% and 24.5% reduction in fruit and vegetable consumption, respectively. African American participants reported a 23.7% and 22.5% decrease in their consumption of lean protein and dairy, respectively. Additionally, 36.9% of the Asian sample indicated a decline in grain consumption since the COVID-19 pandemic (Table 3).

### 3.3. Comparison of Nutritional Risk before and since COVID-19

Table 4 presents a comprehensive overview of the nutritional risk of different racial groups before and since the onset of the COVID-19 pandemic. Before the pandemic, the data revealed that a small percentage of each racial group fell within the “not at risk” category, with African American individuals at 5.31%, Hispanic individuals at 5.83%, White individuals at 6.90%, and Asian individuals at 9.27%. However, with the exception of the White population, there was a slight increase in these percentages since the pandemic. For the “possible risk” category, before the pandemic, the percentages were as follows: African American individuals at 34.96%, Asian individuals at 43.65%, Hispanic individuals at 32.17%, and White individuals at 36.67%. Since the pandemic, these percentages saw a slight decrease, with the exception of the Hispanic population, which experienced a slight increase.

Critically, the majority of the individuals in each racial group were categorized as “at risk” before and since the pandemic. The Hispanic population had the highest percentage at 62% before the pandemic, followed by African American at 59.73%, White at 56.43%, and Asian at 47.08%. Since the pandemic, there was an increase in the “at risk” category for African American (3.3%), Asian (2.42%), and White individuals (2.01%), while only the Hispanic population exhibited a decrease in this category (−1.39%). In this series of analyses, the *p*-values before and since the COVID-19 pandemic for all levels of nutritional risk were <0.001.

### 3.4. Logistic Regression Results for Food Consumption Changes

The results of the binary logistic regression model are reported in Table 5. The results revealed that race was a significant independent predictor of lower consumption of fruit (*p* = 0.009), grains (*p* = 0.001), lean protein (*p* < 0.001), dairy (*p* < 0.001), FSS (*p* < 0.001), and processed meat (*p* < 0.001) since the COVID-19 pandemic (Table 5).

Fruit: African American and Hispanic individuals were 1.21 times and 1.25 times more likely, respectively, to report decreased fruit consumption since the pandemic compared to White individuals (*p* = 0.003 and *p* = 04). However, this decrease was not statistically significant for other racial pair comparisons (*p* > 0.05).

Grains: African American and Asian individuals exhibited a significant decrease in grain consumption since the pandemic (OR = 1.71, *p* = 0.015 and OR = 1.34, *p* = 0.04, respectively) compared to White individuals. Hispanic individuals did not show significant changes in grain consumption compared to any other race.

Vegetables: Hispanic individuals were 1.28 times more likely to report decreased vegetable consumption since the pandemic compared to White individuals (*p* = 0.04). Other racial groups did not exhibit statistically significant changes.

Lean protein: African American individuals had a higher likelihood of reduced lean protein consumption compared to their White counterparts since the pandemic (OR = 1.36, *p* < 0.001). Interestingly, Asian individuals were 23.3% less likely to consume lean protein than African American individuals (*p* = 0.03). Other racial comparisons of lean protein consumption since COVID-19 were not statistically significant.

Dairy: African American individuals, compared to their White counterparts, were more likely to consume less dairy (OR = 1.43, *p* < 0.001) since the pandemic. Hispanic individuals also showed a notable decrease in dairy consumption compared to White individuals (OR = 1.34, *p* = 0.02) since the pandemic. Asian populations were 36.6% and 32.6% less likely to have reduced their dairy consumption when compared to African American and Hispanic individuals, respectively (*p* < 0.001, *p* = 0.02).

FSS: Compared to White individuals, African American individuals reported a notable and statistically significant decrease in FSS consumption (OR = 1.15, *p* = 0.02). In contrast, Asian individuals demonstrated a significant decrease in FSS consumption compared to White individuals (OR = 1.37, *p* = 0.03).

Processed meats: African American individuals were significantly more likely to consume less processed meat since the COVID-19 pandemic compared to White individuals (OR = 1.43, *p* < 0.001). In addition, Asian populations were 39.9% and 31.5% less likely to consume less processed meat in comparison to African American and Hispanic individuals, respectively (*p* < 0.001 and *p* = 0.03).

### 3.5. Racial Disparities in Nutritional Vulnerability

Subsequent binary logistic regression analysis showed that race had a statistically significant association with the likelihood of being “nutritionally vulnerable” following COVID-19 (*p* = 0.02). Based on the results, African American individuals had a statistically significant higher likelihood of being “nutritionally vulnerable” since COVID-19 in comparison to White or Asian individuals (OR = 1.35, *p* = 0.003 for White; OR = 1.38, *p* = 0.04). After controlling for the other variables in the model, African American individuals had a 35% and 38% higher likelihood of being “nutritionally vulnerable” since COVID-19 compared to White and Asian participants, respectively. There were no significant differences between comparisons across other races. All of the details are reported in Table 6.

## 4. Discussion

The COVID-19 pandemic has had a lasting impact on many facets of daily life, notably altering dietary habits and nutritional intake. This study examined the evolving dietary changes of adults from four racial backgrounds during the later stages of the pandemic. What sets this study apart is its comprehensive approach, encompassing all MyPlate food categories in conjunction with FSS and processed meats. The findings of the study also carry significant weight, as they provide valuable insights into how the pandemic affected nutritional risk, particularly in relation to individuals becoming nutritionally “at risk” following the pandemic. These results underscore the importance of paying closer attention to specific food groups during future health crises to safeguard vulnerable populations from potential harm.

Based on the logistic regression analysis findings, race emerged as a significant independent predictor of the reduced consumption of all MyPlate items, FSS, and processed meats among the study participants, with the sole exception of vegetable consumption. These findings highlight the significant dietary disparities among individuals from diverse racial backgrounds, aligning with prior research highlighting consistent disparities in dietary intake across various racial and ethnic groups [37]. In addition, the reduced consumption of MyPlate items, especially fruit, grains, lean protein, and dairy products, can have adverse health implications. The lower intake of these essential food groups can contribute to nutritional deficiencies, increase the risk of chronic diseases, and impact overall well-being [38].

The findings of this study revealed a decrease in fruit intake among African American individuals of 21% and Hispanic individuals of 25% compared to White individuals since COVID-19. This decline might be attributed to several factors, including socioeconomic disparities, cultural preferences, and dietary habits [39,40]. In addition, grain consumption since COVID-19 decreased by 17% and 34% among African American and Asian individuals, respectively, compared to White participants. The United States Department of Agriculture found a significant increase of over 3% in the price of cereal during the pandemic [41]. Accordingly, the reduced consumption of grains could be a result of a decreased purchasing ability, particularly among African American individuals. This finding also raises important questions about the potential long-term impact on dietary patterns; for example, a study by Liu et al. (2023) predicted that grain demand in China would drop by between 5.9% and 11% by 2025 to adopt the balanced diet suggested by the Dietary Guidelines for Chinese Residents 2022 [42].

Another noteworthy observation is the 28% higher likelihood of Hispanic individuals experiencing a decrease in vegetable consumption compared to White individuals since COVID-19; other racial groups did not exhibit significant differences in vegetable consumption changes. It is important to note that mean vegetable consumption before and since COVID-19 was lower among Hispanic individuals in comparison to other racial groups. Crucially, reduced vegetable intake raises concerns about weight gain and heightened risks of poor metabolic health and chronic diseases [43,44].

Further disparities emerged in protein consumption, with African American individuals being 23.3% more likely to consume low amounts of lean protein compared to Asian individuals since COVID-19. Asian individuals’ comparatively higher likelihood of consuming lean protein may stem from cultural dietary preferences, access to certain foods, and traditions favoring specific protein sources [45]. Our finding aligns with a previous study by Beasley (2020), who found that Asian individuals generally had a higher protein intake as a percentage of their calories compared to White and African American populations [46]. Another study by Monroe-Lord, Ardakani, and Spechler (2022) examined the consumption of chicken/turkey and fish/seafood as a source of lean protein and found a significant reduction in the consumption of all of these foods after COVID-19 among African American and White participants; Asian participants, meanwhile, did not significantly decrease their lean protein consumption [47]. Food insecurity, which disproportionately affects African American households, may have played a role in the protein source shifts observed in this study. The increase in food prices, particularly for meats, processed meats, and dairy products during the pandemic, could have had a more pronounced impact on this group, limiting their access to these food items and contributing to the significant reductions in their consumption compared to other racial groups [28,41]. The difference in findings may be attributed to the focus on a limited selection of lean protein sources and the aggregate consideration of dietary intake.

Similarly, the impact of the COVID-19 pandemic on dairy habits also varied among racial and ethnic groups. African American and Hispanic individuals were more likely to reduce their dairy consumption in comparison to their White counterparts, while Asian populations displayed relative resilience in maintaining their dairy intake throughout. This finding aligns with a prior study by Monroe-Lord, Spechler, and Ardakani (2022), who reported a decrease in milk consumption among non-White participants compared to White participants since the onset of the COVID-19 pandemic [48].

In terms of FSS consumption, Asian and African American individuals were 37% and 15% more likely, respectively, to reduce their intake since COVID-19 compared to White populations, suggesting a shift toward healthier dietary practices after the pandemic [4,13].

Significant disparities were observed in the findings regarding processed meat consumption across racial groups since the COVID-19 pandemic. African American participants were more likely to have reduced their intake of processed meats compared to White participants. This suggests that public health messages regarding the risks associated with processed meats may have had greater resonance within African American communities. Alternatively, this group may have experienced more significant changes in food access or purchasing patterns. Conversely, Asian individuals were less likely to reduce their processed meat consumption compared to both African American and Hispanic individuals. These differences are noteworthy, considering the health implications of processed meat consumption. Research indicates that higher intake of processed meat is linked to an increased risk of cognitive disorders such as dementia and Alzheimer’s disease. A cohort study found a significant association between meat consumption and the risk of incident dementia [49]. Furthermore, processed meat consumption is associated with a higher risk of developing type 2 diabetes mellitus [50]. These varied patterns of consumption underscore the complex interplay of cultural, socioeconomic, and environmental factors influencing dietary choices during a public health crisis.

Cultural preferences and differences in food purchasing and preparation practices across racial and ethnic groups in the United States may have influenced the dietary shifts observed in this study. African American, Hispanic, Asian, and White populations tend to have distinct culinary traditions and access to different food types, which likely contributed to the variations in food consumption patterns, particularly during a time of food shortages and price increases. Furthermore, the findings demonstrate a meaningful relationship between race and being “nutritionally vulnerable” following the COVID-19 pandemic; that is, certain racial groups were more likely to experience nutritional vulnerability since the pandemic. The findings of the study suggest that being African American, in particular, is a significant factor associated with higher nutritional vulnerability since COVID-19 compared to White and Asian populations, hinting at underlying socioeconomic, systemic, and health-related disparities that may have been exacerbated by the pandemic [26]. Conversely, Asian participants’ relatively lower risk when compared to African American individuals may point toward possible differences in community support systems, dietary habits, or the impacts of the pandemic between these groups [37].

A notable strength of this study is its robust and comprehensive analysis of dietary changes due to COVID-19 across diverse racial groups, utilizing a large sample size of over 10,000 urban U.S. adults aged 40 to 100. Its methodical use of the validated DST provided an in-depth assessment of participants’ nutritional status, while its focus on racial disparities offered critical insights into the unique dietary impacts and vulnerabilities of these populations during the latter stages of the pandemic. Furthermore, the study’s statistical rigor, particularly its use of binary logistic regression models, lends credence to its findings on consumption trends and nutritional risks. The research not only contributes significantly to the body of nutritional epidemiology literature but also holds substantial implications for the development of culturally tailored public health interventions and policies aimed at mitigating adverse dietary changes and promoting healthy eating practices across diverse communities.

The limitations of the study may include its cross-sectional design, which does not allow for causal inferences between COVID-19 and changes in dietary patterns. Another limitation could be the reliance on self-reported data, which may be subject to recall bias or social desirability bias, potentially affecting the accuracy of the reported dietary changes. There may also be limited generalizability due to the focus on urban populations, which might not reflect the experiences of rural communities. Additionally, the study’s timeframe only captured a specific phase of the pandemic, which may not account for evolving dietary behaviors over time. One more limitation is that this study focuses primarily on dietary changes without considering broader contextual factors, such as physical activity, mental health, or social isolation, all of which could have influenced eating behaviors during the COVID-19 pandemic. Finally, while the study examined racial disparities, it may not fully capture the nuances of socioeconomic factors, access to food, or individual health beliefs that can influence dietary habits. Additionally, it did not consider the full scope of cultural diversity within each racial group.

The future directions of dietary studies post-COVID-19 should focus on longitudinal research to ascertain causal links and trends over time, encompass diverse demographic samples, and examine the effectiveness of public health interventions. While understanding the role of race in such dynamics is important, it is also critical to explore the sociocultural and economic factors influencing food choices, the impact of technological tools in dietary management, and the subsequent health outcomes of pandemic-induced dietary shifts. This knowledge will be vital to developing robust strategies to promote and sustain healthy eating behaviors during global health crises.

## 5. Conclusions

In conclusion, this study provided a crucial examination of the racial differences in dietary changes since the COVID-19 pandemic, revealing significant shifts in food consumption patterns that suggest increased nutritional vulnerability among African American individuals compared to other racial groups. The findings underscore the urgent need for culturally sensitive and equitable public health strategies and interventions that consider the specific challenges and dietary behaviors of diverse populations. Addressing these disparities is imperative to ensure that all communities have the resources and support needed to maintain healthy dietary practices in the face of ongoing and future public health crises.

## Figures and Tables

**Table 1 nutrients-16-03164-t001:** Demographic characteristics of participants.

Characteristic	Frequency (*n*)	Percentage (%)
**Gender**		
Male	4283	42.62
Female	5767	57.38
**Age (years)**		
40–60	3866	38.47
61–80	5908	58.79
81–100	263	2.62
Missing	13	0.13
**Ethnicity/race**		
White	7390	73.53
African American	1393	13.86
Asian	701	6.98
Hispanic	429	4.27
Missing	137	1.36
**Education**		
Less than high school	1634	16.26
Some college	3210	31.94
College degree and more	5191	51.65
Missing	15	0.15
**Annual income (USD)**		
Less than 25,000	1551	15.43
25,000–49,999	2243	22.32
50,000–99,999	3281	32.65
100,000 or more	2566	25.53
Missing	409	4.07

**Table 2 nutrients-16-03164-t002:** Changes in the mean scores and consumption of different food items since the COVID-19 pandemic by race.

Food Item	Race	Before	Since	MPC	*p*-Value
		Mean (SD)	Mean (SD)	(%)	
Fruit	African American	8.66 (3.64)	8.25 (3.79)	−4.76	<0.001
	Asian	8.73 (3.32)	8.43 (3.45)	−3.41	<0.001
	Hispanic	8.43 (3.51)	7.83 (3.73)	−7.15	<0.001
	White	8.43 (3.75)	8.1 (3.88)	−3.97	<0.001
Grains	African American	7.16 (4.3)	6.62 (4.36)	−7.54	<0.001
	Asian	7.44 (4.64)	6.74 (4.61)	−9.39	<0.001
	Hispanic	6.93 (4.15)	6.37 (4.15)	−8.00	<0.001
	White	7.61 (4.56)	7.06 (4.63)	−7.16	<0.001
Vegetables	African American	8.52 (4.03)	8.49 (4.18)	−0.31	0.88
	Asian	8.85 (4.11)	8.88 (4.22)	0.30	0.87
	Hispanic	7.93 (4.35)	7.9 (4.63)	−0.35	0.96
	White	8.22 (4.12)	8.25 (4.2)	0.40	0.045
Lean protein	African American	5.64 (2.49)	5.48 (2.53)	−2.76	0.003
	Asian	5.24 (2.64)	5.19 (2.8)	−0.93	0.42
	Hispanic	5.03 (2.36)	4.9 (2.51)	−2.58	0.18
	White	5.13 (2.42)	5.08 (2.49)	−0.98	0.03
Dairy	African American	4.18 (2.58)	4.03 (2.57)	−3.43	0.004
	Asian	4.2 (2.66)	4.18 (2.73)	−0.64	0.58
	Hispanic	4.7 (2.53)	4.5 (2.55)	−4.34	0.02
	White	4.57 (2.56)	4.53 (2.59)	−0.84	0.006
Fat, sugar, and sweets	African American	13.39 (4.63)	14 (4.7)	4.56	<0.001
	Asian	15.12 (4.27)	15.83 (4.45)	4.64	<0.001
	Hispanic	14.22 (4.64)	14.99 (4.73)	5.36	<0.001
	White	13.56 (4.41)	13.99 (4.64)	3.18	<0.001
Processed meats	African American	6.34 (2.98)	6.44 (2.85)	1.62	0.07
	Asian	8.14 (2.3)	8.17 (2.28)	0.35	0.81
	Hispanic	6.87 (2.82)	6.91 (2.58)	0.50	0.87
	White	7.29 (2.62)	7.24 (2.6)	−0.67	0.02

MPC: mean percentage change.

**Table 3 nutrients-16-03164-t003:** Racial variations in food item consumption changes since the COVID-19 pandemic.

Food Item	Race	Decreased Consumption	No Change	Increased Consumption	*p*-Value
		N (%)	N (%)	N (%)	
Fruit	African American	501 (35.97)	588 (42.21)	304 (21.82)	
	Asian	224 (31.95)	336 (47.93)	141 (20.11)	
	Hispanic	160 (37.30)	186 (43.36)	83 (19.35)	
	White	2215 (29.97)	3918 (53.02)	1257 (17.07)	
	*p*-value				<0.001
Grains	African American	492 (35.32)	611 (43.86)	290 (20.82)	
	Asian	259 (36.95)	338 (48.22)	104 (14.84)	
	Hispanic	151 (35.20)	186 (43.36)	92 (21.45)	
	White	2285 (30.92)	4047 (54.76)	1058 (14.32)	
	*p*-value				<0.001
Vegetables	African American	287 (20.60)	795 (57.07)	311 (22.33)	
	Asian	144 (20.54)	405 (57.77)	152 (21.68)	
	Hispanic	105 (24.48)	239 (55.71)	85 (19.81)	
	White	1245 (16.85)	4746 (64.22)	1399 (18.93)	
	*p*-value				<0.001
Lean protein	African American	330 (23.69)	800 (57.43)	263 (18.88)	
	Asian	126 (17.97)	460 (65.62)	115 (16.41)	
	Hispanic	82 (19.11)	279 (65.03)	68 (15.85)	
	White	1206 (16.32)	5140 (69.55)	1044 (14.13)	
	*p*-value				<0.001
Dairy	African American	313 (22.47)	819 (58.79)	261 (18.74)	
	Asian	103 (14.96)	481 (68.62)	117 (16.69)	
	Hispanic	91 (21.21)	267 (62.24)	71 (16.55)	
	White	1111 (15.03)	5223 (70.68)	1056 (14.29)	
	*p*-value				<0.001
Fat, sugar, and sweets	African American	595 (42.71)	436 (31.30)	362 (25.99)	
	Asian	321 (45.79)	240 (34.24)	140 (19.97)	
	Hispanic	192 (44.76)	143 (33.33)	94 (21.91)	
	White	2794 (37.81)	2883 (39.01)	1713 (23.18)	
	*p*-value				<0.001
Processed meats	African American	284 (20.39)	852 (61.16)	88 (12.55)	
	Asian	91 (12.98)	522 (74.47)	257 (18.45)	
	Hispanic	82 (19.11)	256 (59.67)	91 (21.21)	
	White	986 (13.34)	5255 (71.11)	1149 (15.55)	
	*p*-value				<0.001

**Table 4 nutrients-16-03164-t004:** Nutritional risk assessment among racial groups before and since the COVID-19 pandemic.

		African American	Asian	Hispanic	White
		N (%)	N (%)	N (%)	N (%)
Not at risk	Before	74 (5.31)	65 (9.27)	25 (5.83)	510 (6.90)
	Since	80 (5.74)	77 (10.98)	26 (6.06)	485 (6.56)
Possible risk	Before	487 (34.96)	306 (43.65)	138 (32.17)	2710 (36.67)
	Since	435 (31.23)	277 (39.51)	143 (33.33)	2586 (34.99)
At risk	Before	832 (59.73)	330 (47.08)	266 (62.00)	4170 (56.43)
	Since	878 (63.03)	347 (49.50)	260 (60.61)	4319 (58.44)

**Table 5 nutrients-16-03164-t005:** Comparative odds ratios for changes in food consumption since COVID-19 among various racial groups.

	Odds Ratio	95% CI ^1^	*p*-Value
**Fruit**			
			0.009
African American vs. White	1.21	1.07, 1.38	0.003
Asian vs. White	1.09	0.91, 1.29	0.35
Hispanic vs. White	1.25	1.01, 1.54	0.04
Asian vs. African American	0.90	0.73, 1.10	0.29
Asian vs. Hispanic	0.87	0.67, 1.13	0.30
African American vs. Hispanic	0.97	0.77, 1.22	0.80
**Grains**			
			0.001
African American vs. White	1.17	1.03, 1.33	0.01
Asian vs. White	1.34	1.13, 1.59	<0.001
Hispanic vs. White	1.18	0.95, 1.46	0.13
Asian vs. African American	1.15	0.94, 1.40	0.17
Asian vs. Hispanic	1.14	0.88, 1.48	0.32
African American vs. Hispanic	0.99	0.79, 1.25	0.95
**Vegetables**			
			0.21
African American vs. White	1.06	0.91, 1.24	0.45
Asian vs. White	1.08	0.87, 1.32	0.50
Hispanic vs. White	1.28	1.01, 1.63	0.04
Asian vs. African American	1.01	0.80, 1.29	0.91
Asian vs. Hispanic	0.84	0.62, 1.13	0.25
African American vs. Hispanic	0.82	0.64, 1.07	0.15
**Lean protein**			
			<0.001
African American vs. White	1.36	1.17, 1.58	<0.001
Asian vs. White	1.04	0.84, 1.29	0.69
Hispanic vs. White	1.07	0.83, 1.39	0.58
Asian vs. African American	0.77	0.60, 0.98	0.03
Asian vs. Hispanic	0.97	0.71, 1.33	0.85
African American vs. Hispanic	1.26	0.96, 1.67	0.09
**Dairy**			
			<0.001
African American vs. White	1.43	1.22, 1.66	<0.001
Asian vs. White	0.90	0.72, 1.14	0.39
Hispanic vs. White	1.34	1.05, 1.72	0.02
Asian vs. African American	0.63	0.49, 0.82	<0.001
Asian vs. Hispanic	0.67	0.49, 0.93	0.01
African American vs. Hispanic	1.06	0.81, 1.39	0.66
**Fat, sugar, and sweets**			
			<0.001
African American vs. White	1.15	1.02, 1.30	0.02
Asian vs. White	1.37	1.16, 1.62	0.03
Hispanic vs. White	1.25	1.02, 1.53	0.50
Asian vs. African American	1.19	0.98, 1.44	0.07
Asian vs. Hispanic	1.10	0.85, 1.41	0.47
African American vs. Hispanic	0.92	0.74, 1.15	0.47
**Processed meats**			
			<0.001
African American vs. White	1.43	1.22, 1.67	<0.001
Asian vs. White	0.86	0.68, 1.09	0.22
Hispanic vs. White	1.25	0.97, 1.63	0.09
Asian vs. African American	0.60	0.46, 0.79	<0.001
Asian vs. Hispanic	0.68	0.49, 0.96	0.03
African American vs. Hispanic	1.14	0.86, 1.51	0.36

^1^ CI: Confidence interval.

**Table 6 nutrients-16-03164-t006:** Comparative odds ratios for state of being more nutritionally vulnerable since the COVID-19 pandemic by race.

	Odds Ratio	95% CI ^1^	*p*-Value
More Nutritionally Vulnerable			
			0.02
African American vs. White	1.35	1.11, 1.64	0.002
Asian vs. White	0.97	0.73, 1.28	0.84
Hispanic vs. White	0.92	0.64, 1.33	0.66
Asian vs. African American	0.72	0.52, 0.99	0.04
Asian vs. Hispanic	1.05	0.68, 1.60	0.81
African American vs. Hispanic	1.46	0.99, 2.16	0.056

^1^ CI: Confidence interval.

## Data Availability

Data used during the current study are available from the corresponding author.

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
