# Peer review of "Dietary Shifts since COVID-19: A Study of Racial Differences"

_nutrients, 2024, doi:10.3390/nu16183164_

Round 1

Reviewer 1 Report

Comments and Suggestions for Authors

The study aims to assess the impact of the pandemic on nutritional vulnerability across different racial groups, particularly identifying disparities and providing insights into public health strategies. The emphasis on racial and ethnic differences in dietary habits adds valuable insights into nutritional inequities, an area with significant public health implications.

Some comments to improve the paper:

Introduction

The introduction could benefit from a more detailed discussion of the socioeconomic factors that influence dietary behaviors across racial groups.

What were the research hypotheses? Please add.

Methods

It is important to provide information on the psychometric properties of the DST.

Results

Add subtitles to the results section.

Discussion and Conclusions

Expand the discussion of the practical significance of the findings. The conclusions could be more explicit in linking the dietary shifts to potential policy interventions. The discussion on the broader implications for public health policies could be expanded.

Limitations

Please add that the study focuses primarily on dietary changes without considering broader contextual factors such as physical activity, mental health, or social isolation, all of which could have influenced eating behaviors during the pandemic.

Author Response

The response to reviewers has been attached.

Reviewer 2 Report

Comments and Suggestions for Authors

Interesting manuscript, which shows from the optima of racial differences changes in dietary pattern during the pandemic, and although it seems to me a well written manuscript and with clear results, although I have 2 doubts:

1. although it mentions it superficially in the discussion, these results may be associated to the income level of the different ethnic groups, was there any control of that aspect?

2. Is it possible that African Americans were more food insecure? and the increase in the value of meats, processed meats and dairy products affected them more than the other groups?

3. is there any evidence that in the United States, depending on the ethnic group, food preparations or purchases are different and that this is influencing the results?

Author Response

The response to the reviewer has been attached
